# “They Do Not Care about Us Anymore”: Understanding the Situation of Older People in Ghana

**DOI:** 10.3390/ijerph18052337

**Published:** 2021-02-27

**Authors:** Joseph Asumah Braimah, Mark W. Rosenberg

**Affiliations:** Department of Geography and Planning, Queen’s University, Kingston, ON K7L 3N6, Canada; mark.rosenberg@queensu.ca

**Keywords:** Ghana, older people, lived experiences, sharing circles, ecological systems theory

## Abstract

While existing research acknowledges copious challenges faced by older adults (people aged 60 and over) in Ghana and most countries in sub-Saharan Africa, they fail to situate the lived experiences of this vulnerable group within the broader context of health geography and public health. This paper draws insights from ecological systems theory and the “geographies of older people” literature to examine the lived experiences of older people in Ghana. Data for the study were gathered using interviews (42) and sharing circles (10). Our findings reveal a complex mix of experiences consistent with the different levels of the environment. Dominant themes include access to social support, functional impairment and poor health status, social status, poor access to water and sanitation services, food insecurity, economic insecurity, and caregiving burden. These findings support the wide-held notion that the experiences of older people are complex and produced by the interplay of both individual and structural factors. Our findings demonstrate that sociocultural, economic, political, and climatic factors are important consideration in promoting elderly wellbeing and quality of life in Ghana.

## 1. Introduction

Estimates show that the number of people aged 60 and over is increasing globally. Specifically, in 2017, the older population numbered 962 million, an increase from 382 million in 1980 [1]. Projections are that by 2030 and 2050, the older population will number 1.41 billion and 2.1 billion respectively [1]. Roughly two-thirds of older people reside in countries in the less developed regions of the world including sub-Saharan Africa (SSA). Not only are the economies of the countries of SSA inadequately positioned to contain this demographic tide, but very little attention has been paid to issues of older people in this context. Promoting elderly wellbeing is key to human development and has been consistently featured in various development agendas such as the Programme of Action of the International Conference on Population and Development (ICPD), the Millennium Development Goals, and quite recently the 2030 Agenda for Sustainable Development [1,2]. 

Old age is characterized by an intricate mix of relationships and conditions. Older people compared to other population categories experience elevated levels of morbidity and functional impairment that affects their ability to live life meaningfully [1,3]. In the specific case of SSA marked by immense health care access barriers, evidence reveal high burden of chronic diseases and modifiable health risk factors among older adults in the region [3,4,5]. Furthermore, older people compared to other age cohorts are at a much higher risk of poverty, loneliness, discrimination, abuse, and stigma [3,6]. Despite these challenges, a considerable proportion of older people still remain economically active and employed, contributing significantly to economic growth [3,4]. More so, older people, remain a critical social support pillar, especially in caring for grandchildren and other vulnerable members in their communities [4,7]. 

Previous empirical evidence attributes the situation of older people to a complex interaction between individual- and structural-level factors [8,9,10,11,12]. For instance, the functional and health status of older people determines their level of independence which in turn shapes their overall wellbeing [12]. Also, household composition, and the nature of social support structures determine resource availability and support for older people [4,11,13,14]. This is particularly critical in settings like Ghana with inadequate formal support systems along with weakening social ties, declining household sizes, and outmigration of adult children [13,15]. Other important factors that define older adults’ conditions are economic status [16], available social security schemes [17], and geographic location [18]. However, the role of geography in constructing the everyday experiences of older people has been paid little attention in the literature which has implications for knowledge and efforts at promoting the wellbeing and quality of life of older people.

Indeed, over the past few decades, human geographers have called to attention the importance of place and space in aging and old age outcomes [19,20,21]. Geographers’ engagement with questions of aging is done under the sub-fields of “geographies of aging” or “geographies of older people” and on the premise of a mutual and complex relationship between aging and the spaces and places within which it occurs [21,22]. Thus, experiences of old age including the coping strategies adopted are produced through complex interactions between structural and individual factors such as place of residence, available support networks, gender, cultural norms, and values, economic conditions, and the policy environment. Geography provides a useful and broad framework to examine and interpret the complex experiences of older people [19,22]. 

Whereas geographic inquiry into aging is evident, and dominant in the global north, there is little engagement with the sub-discipline by researchers in SSA, particularly Ghana. In a context delineated by considerable socioeconomic, cultural, and environmental disparities, employing a geographic lens aids in contextualizing the complex and developing experiences of older people. The rapidly changing Ghanaian social and economic systems add an additional layer of complexity to the circumstances of older adults, thereby meriting a geographic lens. This study contributes to knowledge and policy practice by situating the everyday experiences of older adult Ghanaians within the broader “geographies of older people” framework. Ghana’s older adult population has increased rapidly over the years. According to Ghana’s recent population and housing census, people aged 60 and over constituted 6.7% of the country’s population in 2010, a rise from 4.5% in 1960 and is projected to double by 2050 [8]. With Ghanaians now living longer, it is important to pay attention to the situation of older people, especially within research and policy spheres. This will allow for the efficient allocation of resources for national development.

## 2. Aging and Ecological Systems Theory

Ecological perspectives which have gained considerable research and policy attention over the past three decades in the behavioral sciences assume that behavior has multiple levels of influence and that these influences interact across distinct levels [23,24]. In line with this, ecological perspectives emphasize the need for multi-level interventions in changing human behaviors. In this study we adopt the ecological systems theory proposed by Bronfenbrenner to situate the lived experiences of older people in their social and physical contexts. This theoretical construct allows for a broader interrogation and understanding of the human environment as an agent of development. Bronfenbrenner views the environment as “a nested arrangement of structures, each contained within the next” [25] (pp. 514). Based on this conception, five levels of the human environment (the microsystem, the mesosystem, the exosystem, the macrosystem and the chronosystem) need to be considered. 

Drawing from this perspective, we view aging as a complex process affected by and affecting different levels of the social and physical environments. At the center of the theory is the developing person, which in the context of this study is the growing older person and their personal characteristics. The microsystem, the first level comprises the older person and their immediate environment and all interactions between them. This includes older people’s physical surroundings such as the home and workplace. Previous studies in Ghana and beyond have highlighted the usefulness of microsystem factors in constructing the experiences of older people [26,27]. A mesosystem involves the interconnections among the major settings in which older people are found at any point in their life. Among older people, their mesosytem include interactions among family, peers, church, and workplace. The exosystem is an extension of the mesosystem and involves structures that do not themselves contain developing older persons, but impact the settings containing them. These structures can be formal or informal and include neighborhoods, mass media, government institutions, and government policies. Amegbor and colleagues draw attention to the relevance of neighborhood environment in defining the mental wellbeing of older people in Ghana [11]. They found neighborhood safety and structural social capital to reduce the likelihood of depression among older adults. The macrosystems encompass the “overarching institutional patterns of the culture or subculture, such as the economic, social, educational, legal, and political systems, of which micro-, meso-, and exo- systems are the concrete manifestations” [25] (pp. 515). The macrosystem borders on the ideologies of a group that shape behaviors and relationships. The chronosystem, a level added later involves all the experiences of individuals over the course of their lives. 

The complexities of aging require a theoretical perspective that considers the different and interconnected influences of the environment. Ecological systems theory coupled with perspectives from the “geographies of older people” therefore provides a theoretical advantage in situating the everyday experiences of older people within prevailing social, economic, political, and environmental contexts. These frameworks also provide the positional space to develop and implement appropriate intervention programs to improve the wellbeing of older people in Ghana. Earlier researchers have applied ecological systems theory to investigate various social phenomena among vulnerable groups in Ghana. For instance, Yendork used ecological systems theory to explore the different vulnerabilities associated with orphanhood in the Ghanaian context [28]. Also, Agyei-Okyere and colleagues drew on ecological systems theory to examine the living conditions of disabled people in mining communities, concluding that participants’ interactions with the environment produced unbearable hardships [29]. However, no study to the best of our knowledge has utilized this perspective in exploring the experiences of older people in Ghana.

## 3. Materials and Methods

### 3.1. Study Design

This study employed a qualitative research design, involving interviews and sharing circles to understand the experiences of older adults in Ghana. Interviewing as a data collection technique explores individuals’ views, experiences, and/or beliefs about a phenomenon [30,31]. Interviewing has been touted for its flexibility and effectiveness in unearthing people’s experiences, and the meanings they attach to these experiences. Sharing circles on the other hand are open conversations involving the sharing of stories and life experiences of people within a particular cultural setting [32,33]. Rooted in indigenous epistemologies and ways of understanding the world, sharing circles have emerged as a vital research tool in conducting research in Indigenous communities within Europe and North America [34]. A combination of these two qualitative data collection techniques is appropriate to unearth the complex experiences of older people in a natural and culturally appropriate manner. 

The consolidated criteria for reporting qualitative research (COREQ) proposed by Tong and colleagues was utilized in reporting our findings [35]. COREQ is a tool developed to aid in the explicit and comprehensive reportage of qualitative findings [35]. 

### 3.2. Research Team and Reflexivity

Interviews were conducted by the lead author, a doctoral student who identifies as a male and hails from the Upper West Region of Ghana. No participant had prior relationship with the lead author who conducted the interviews, with the support of trained research assistants and community leaders. Though in navigating language barriers we recruited research assistants from the area, they had no relationship or contact with participants. The co-author (MWR) is the lead author’s doctoral supervisor who has no contact with the participants. MWR is a human geographer with interests in health and health service use, aging, and population studies among others, employing both qualitative and quantitative research techniques. The assumptions of the researchers going into this study were that old age experiences in Ghana are diverse and produced by the complex interaction of individual and structural factors. This research contributes to the gerontology literature by illuminating the lived experiences of older adults in Ghana. The study further provides evidence for relevant policy decision-making in Ghana and similar contexts. 

### 3.3. Study Setting

The Upper West (Savannah zone), Bono East (The Bono East Region was carved out of the Brong Ahafo Region in 2019. Statistics presented in this manuscript therefore cover the former Brong Ahafo Region.) (Forest zone) and Greater Accra (Coastal zone) Regions of Ghana were selected for this study. These regions were purposively selected to represent the three major ecological zones in Ghana (see Figure 1). Data collection took place from June to August 2019. Ghana’s most recent population census reports that the older population constitutes 6.7% (approximately 1.6 million) of the estimated 25 million people in the country [36]. Of the total number of older people, the Greater Accra Region (12.9%) has the largest proportion, with the Brong Ahafo and Upper West Regions constituting 8.7% and 3.6% respectively [8]. 

Substantial socioeconomic variations exist across these ecological zones, and between urban and rural areas, which are crucial for understanding people’s lived experiences [8,37]. Reasons for these variations are political, environmental, and sociocultural. For example, precolonial and current governmental policies led to more development in the Coastal and Forest zones than the Savannah zone [38]. Poverty incidence in the Greater Accra Region (2.5%) is way below the national average (23.4%), and above average in the Brong Ahafo (26.8%) and Upper West Regions (70.9%) [39]. Doctor to population ratios in 2017 were highest in the Greater Accra Region (3,052), followed by the Brong Ahafo Region (10,059) and the Upper West Region (14,821) [40]. Compared to urban areas, rural areas in Ghana where the majority (54%) of older people reside are underdeveloped and lack basic amenities including health facilities, electricity, potable water, and sanitation services [37]. In addition to outnumbering their male counterparts, possibly due to their higher life expectancies, older women (56%) are socioeconomically disadvantaged and continue to play productive and reproductive roles including providing care for grandchildren and other vulnerable household members [3,37]. 

Outmigration is a common phenomenon in Ghana, involving mostly younger adults in rural areas and the Savannah ecological zone in search of better livelihood opportunities [41]. Religion forms an integral part of life in Ghana with 9 out of 10 older Ghanaians being religious. Over half (58.5%) of Ghana’s older population are economically active, constituting an estimated 8.6% of the country’s labor force [8]. An overwhelming majority (96%) of these economically active old people are employed; predominantly in the agricultural and forestry sectors (85%) [8]. A little over half of the older population are married (53.2%), illiterate (60%), and household heads (62%); with 12% of them being disable [8]. About a quarter (22%) of houses occupied by older people do not have toilet facilities [8]. 

In July 2010, Ghana adopted a policy on aging to bring to the fore issues confronting the older population [8,17]. Additionally, the National Health Insurance Scheme (NHIS) and the Livelihood Empowerment Against Poverty (LEAP) program, recognizing the needs of the older population, provide free medical insurance and cash grants to some needy older people in the country [6,42]. These policy interventions have however been decried for being inadequate and incomprehensive [6]. 

The study communities chosen in the context of the national and regional situation were La Nkwantang Municipal and Ningo-Prampram District in the Greater Accra Region, Techiman and Nkoranza South Municipalities in the Bono East Region, and Wa West and Sissala East Districts in the Upper West Region (see Figure 1).

### 3.4. Participants

Participants for the study were Ghanaians aged 60 and over. Age 60 was adopted by the central government as the age of retirement, hence the benchmark for defining an old person in Ghana. Our participants were drawn to represent diverse socioeconomic and geographical backgrounds (such as education, rural/urban areas, ecological zones, and gender). This was done to ensure varied old age experiences were uncovered. We excluded older people who were frail and/or cognitively incapable of participating in the study. 

### 3.5. Sampling and Recruitment

A purposive sampling technique was used to recruit research communities and participants. Purposive sampling involves the selection of participants because they possess certain characteristic necessary for detail exploration of a particular theme or research question. This ensures that all relevant participants are included in the study which allows for the exploration of diverse perspectives on a topic. Interviews commenced with contacting gatekeepers of selected communities, who in most cases were assembly members. The gatekeepers then assisted the research team to identify and contact potential participants. The scope of the study was explained to participants, and interviews scheduled. Six (6) participants, however, could not make it to the interview due to ill-health and other personal reasons. In situations where a participant could not participate, efforts were made to find a replacement with similar characteristics. 

The sharing circles also commenced with contacting the gatekeepers of selected communities, who were subsequently briefed on the study. We then asked for their assistance to recruit participants. We also sought information on knowledge acquisition protocols in the communities from these gatekeepers. Follow-up visits were made a few days later for updates, further consultation, and scheduling. The visits also aided in building rapport with our research participants. Even though group membership was decided by the community, we ensured they fell within the target population. The number of participants in the group was decided by the community. The minimum and maximum number of participants in the sharing circles were 6 and 10, respectively. 

### 3.6. Data Collection

Interviews were conducted face-to-face in Twi, Sissali, Dagaare, Brifo, Ga, or English at participants’ preferred locations. Through this, participants were comfortable sharing their stories and experiences. Interviews were semi-structured. Semi-structured interviewing is a data collection technique with a predetermined order, that also allows for divergence to explore an idea in detail [43]. Our interview guide comprised open-ended questions. The guide was developed by the research team based on the research objectives, comprising two main sections. The guide was pretested among four participants and refined to fit the study. The first section focused on older people’s understanding of aging. The second part centered on their experiences. The first author conducted all interviews with support from trained research assistants and gatekeepers. The interviews lasted between 20 and 40 min and were audio-recorded with permission from participants. A total of 42 interviews were conducted (19 in Upper West Region, 10 in Bono East Region, and 13 in Greater Accra Region). 

Ten sharing circles were conducted―five each in the Upper West and Bono East Regions. An urban and rural community were selected from each of these two regions for the sharing circles. Three meetings were held in each rural area comprising an all-male, an all-female, and a mixed one. For each urban area we conducted two meetings, comprising an all-male group and an all-female group. We developed a guide for the sharing circles based on the study objectives. A participant was selected from each group to facilitate the discussion. Sharing circles lasted two hours on average and were audio-recorded with permission from the participants. Our sample sizes (interviews and sharing circles) were defined by “information power” [44]. The authors suggest that an adequate sample size for any study is determined by the aim, sample specificity, theoretical approach, quality of dialogue, and analytic strategy. We also collected field notes to reflect on, complement, and contextualize our data. Each participant was given 10 Ghana cedis (approximately 2 United States Dollars) for their time. For some sharing circles, kola nuts were provided in accordance with customs. 

### 3.7. Data Management, Analysis, and Rigour

Interviews were translated and transcribed by the first author and two professional transcribers. For each interview transcript, two other persons proficient in the language were contracted to compare the transcripts with the corresponding audio file. Two other local researchers were asked to randomly sample and compare transcripts with audio recordings. This was done to maintain meaning and ensure rigor. Transcripts were then imported into NVivo™12 qualitative data analysis software for analysis. Following Braun and Clarke the lead author read all the transcripts repeatedly to familiarize himself with the data while noting key themes, which was done both deductively and inductively [45]. We then proceeded to generate codes systematically, after which codes were organized into themes and sub-themes. Finally, the themes and sub-themes were mapped to the different systems of the environment proposed by Bronfenbrenner. Transcripts from each data source were coded by the lead author and subsequently by another researcher to ensure reliability. Differences in coding were discussed and resolved. Over 70% agreement was achieved for both interviews and sharing circles [46]. Triangulation of the findings was done across the two data sets. We facilitated the credibility of this study by consistently using interview guides during the interview process. We also used professional transcribers and cross-checked interview transcripts. Additionally, we used qualitative data analysis software. 

### 3.8. Ethics

Research ethical protocols were duly observed. In terms of procedural ethics, the study received clearance from the Queen’s University General Research Ethics Board (GGEOPL-277-19). Beyond the procedural ethics, culturally defined situational, relational, and exiting ethical protocols were duly followed. Written and thumbprint consents were obtained from participants after they were briefed and had all of their questions answered. Participants were assured of confidentiality and those in the sharing circles admonished to remain confidential about what others had said. To protect participants’ identities, we use pseudonyms in presenting quotes. Key participant identifiers have also been excluded from the results.

## 4. Results

### 4.1. Participant Characteristics 

Participant characteristics are presented in Table 1 and Table 2. Our interview participants (see Table 1) were almost evenly distributed between females and males, and rural and urban areas. Additionally, about half of participants were aged 60–69 (48%), had no education (52%), and with household size below 6 (50%). 

Akin to the interviews, our sharing circles (see Table 2.) participants were almost evenly distributed between females and males. Also, the majority of participants were aged 60–69 (54%), had no education (80%), and lived in rural areas (63%).

### 4.2. Participants’ Experiences

This section presents participants’ experiences. Participants’ pseudonyms, gender (F = Female; M = Male), and region of residence (UWR = Upper West Region; BER = Bono East Region; GAR = Greater Accra Region) are associated with quotes. 

#### 4.2.1. Social Support and Loneliness

Access to support was a recurring theme in the interviews. Consistent with Ghana’s social support landscape, participants’ narrations largely bordered around informal support, which encompasses unpaid assistance from family, friends, neighbors, and other community members. We found divergent support experiences with most participants explaining that they lacked assistance with daily living tasks such as washing, cooking, sweeping, and shopping. In the words of a participant: *“There is no support. I do not have any helper and due to my sickness it’s difficult to perform daily tasks. Even getting someone to help prepare food is a problem. My clothes are always dirty (John, M, UWR).”* More older men than women were concerned about the lack of assistance in carrying out these tasks, and this is potentially a reflection of prevailing discriminatory cultural norms where women are ascribed such domestic roles. A number of reasons were cited for the lack of support. Key among them is the outmigration of adult children. This theme was more common among participants from the northern and middle belts of the country and those in rural areas. A few participants blamed the lack of support on the demise of their adult children and partners. Social and economic transformations were also acknowledged as barriers to social support. A participant had this to say: *“Some changes have occurred. Those days, people sympathized with older people. The younger generations assisted the older ones regardless of who they are. Now it is not the case. They do not care about us anymore (Joe, M, UWR).”*


The general practice and expectation are that older persons are cared for by informal networks, especially adult children. Akin to this expectation, a few rural and female participants admitted receiving support from their networks. As highlighted in this participant’s testimony: *“My children care for me. They can never leave me to go hungry. I carried them in my stomach for several months and took care of them when they were young and it’s their turn now (Akua, F, BER).”* Albeit limited, non-kin also play a crucial role in providing support for older people, especially those lacking assistance from family members. As expressed by a participant: 


*“I get food to eat. Someone [non-kin] gave me yam yesterday. I really do not have a problem with food... In my church, they visit the elderly occasionally to help. …There is another church in this area; every month they bring the elderly together to share the word of God and feed them (Kwame, M, BER).”*


It was observed that intangible support, particularly emotional support was less of a concern to participants. A common issue that imbued discussions on social support was how the lack of assistance negatively affects health and wellbeing. Many participants noted engaging in health compromising activities to survive. For some, they were constantly worried and had sleepless nights about where their next help will come from. The lack of support participants explained had the potential to lead them to their “early graves”. Some participants also reported experiencing loneliness. Experiences of loneliness were common among male participants and those living in urban areas where social cohesion appears weak. When asked about support from formal sources, it was apparent very limited assistance was available as many of them were oblivious of the existence of such services. A number of participants, especially rural folks and those from the Upper West Region reported receiving cash support under the LEAP program. Some participants were also exempted from paying health insurance premiums as part of government’s efforts to improve health care access in Ghana. Participants decried, however, that these support services are limited and highly politicized. 

#### 4.2.2. Social Status

Old age in traditional Ghanaian society is generally associated with improved social status and authority. Some participants reported elevated status and authority at this stage of their lives. Some of them have been conferred leadership roles including household, clan, and spiritual headships. 


*“I am the head of the section and every morning they come to greet. I own the land here and make decisions. Payment of dowry is done in my house and when it comes to sharing resources, I receive first. It is the norm (Musah, M, UWR).”*


These ascribed statuses are also useful in expanding conferees’ social capital. *“As an elder, I am respected, and can ask for assistance from any one in my community. At harvest I receive food stuffs from people. Where I live now was put up for me as the head (Batong, M, UWR).”* Discussions of ascribed status and authority, however, reflects gender and socioeconomic disparities with females and poorer individuals recounting lower social status. They reported frequent discrimination, stigma, and abuse. A participant had this to say: 


*“Some young ones say it to me that I am a witch. I am a witch because all my siblings are dead, and I am the only one left. But I am innocent. I do not know how and what witchcraft is. One of my nephews even left the community because of that (Abena, F, BER).”*


In the words of another participant: *“We are the most abused and disrespected ones in this community. We are seen as liabilities. No one respects us. We get insulted all the time and if you are not careful can even be beaten (Hatong, F, UWR).”*

#### 4.2.3. Functional Restriction and Poor Health and Health Service Access 

A recurring theme in the interviews and sharing circles was poor health and functional impairment. Notable health conditions included body pains, diabetes, high blood pressure, arthritis, and hypertension. These conditions considerably limit their independence and ability to engage in livelihood activities. They were often referred to as “aging sickness.” 


*“Today you feel pains here, tomorrow you feel it somewhere else. You go to the hospital and you are told its menopause. Personally, I am always ill, there are times I could not walk because my legs were hurting. Every month I go to the hospital for check-ups (Ama, F, BER).”*


Although many participants reported seeking formal health care service, numerous access barriers exist. Several participants were not eligible for free health care under the national health insurance program as only individuals that are 70 years and over have coverage. The non-availability of geriatric care services and associated financial implications were also notable barriers to health service use. More concerning was the fact that many of those who reported “aging sickness” were ineligible for financing under the NHIS. The difference that access to health insurance makes and how it differentiates older people in our study is illustrated in the quote below:


*“By the Grace of God this NHIS card has made things a bit easier although it does not cover a lot of the medications. … I do not work due to my age and if I start to feel pains in any part of my body instead of me going to the hospital I would not (Hawa, F, BER).”*


In addition to financial barriers, participants in rural areas expressed concerns about geographical barriers to accessing health care with many of them lamenting having to travel longer distances to access health care services. Others identified discrimination at the health facilities explaining that it prevents them from seeking health care services. The inability to access formal health care services often precipitated self-medication and the use of traditional medicines among participants. 

#### 4.2.4. Food Insecurity

Food insecurity was another concern among participants, particularly those in the Upper West and Bono East Regions where livelihoods heavily depend on agriculture. Participants lamented going several days without food and when available, how it was insufficient and lacked relevant nutrients. 


*“Previously I eat the kind of food I want to eat but now I do not have money so whatever my children get for me is what I will eat. At this age it is all about management. Sometimes you can go the whole day without food. You have to wait for them to return from the farm before you can get something to eat (Iddi, M, UWR).”*


Although some older people engage in livelihood activities to meet their nutritional needs in part, returns from these activities are often insufficient. Older people in agriculture for instance complained of climate change threats, high input cost, and low crop yields: 


*“We rely on farming to feed but these days yields are very low. And when that happens your feeding is affected. Inputs are very expensive to come by. Our land is now accustomed to fertilizer. To survive one must reduce the quantity of meals (Abudu, M, UWR).”*


Some participants reported employing food management strategies such as skipping meals, reducing meal quantities, and eliminating certain food groups from their diets. 


*“Just as you mentioned earlier, maybe with the 30 Ghana cedis I have, eating banku will prevent me from getting food tomorrow. Even if I want to take the banku I cannot. Sometimes I go a whole week without taking meat. But the truth is that when you get to a point in life you cannot eat everything you want (Nancy, F, BER).”*


Others begged for food to feed. The lack of consistent access to food was acknowledged to negatively affect their mental and physical health. 

#### 4.2.5. Economic Insecurity

Linked to food insecurity, participants also complained about economic insecurity. Most participants were economically inactive and without pension benefits. Income support from formal institutions was reported to be limited to a few rural and indigent older people. Grant amounts were also noted to be small. An old man on the LEAP program described his cash grant as pitiful. Economic insecurity does not only hinder participants’ access to nutritional diets, but it also impedes health care service use especially among rural dwellers who in many situations have to travel to access health care. 


*“They do not use it (NHIS CARD). During Kufuor’s era (between 2001 and 2008), you do not have to pay for anything when you come to the hospital with the health insurance. But now, when I renew my insurance, I have to pay GHȻ 50.00 (Peter, M, UWR).”*


Many participants lacked collaterals to secure loans. The inability to secure loans considerably affects participants negatively, especially their ability to engage in livelihood activities. A few capable older people in maintaining livelihoods, however, were engaged in some economic activities such as trading, operating food joints, farming, rearing, brewing pito, shea nut picking and processing, charcoal burning and sewing. Undertaking these activities is, however, not without barriers. Commodities, especially food from older people are less patronized. Undertaking these livelihood activities is also physically demanding and health compromising. *“I sew dresses, but my sight is now failing me. Also, as you age patronage declines. You are seen as old fashion (Afia, F, BER).”*


#### 4.2.6. Inadequate Access to Water and Sanitation Services

Experiences with accessing water and sanitation services was also discussed. Akin to the general population, access to water and sanitation services was a challenge to participants. Water access challenges were, however, gendered, with older women and urban dwellers being the worse affected. Rural older woman for instance trekked longer distance to draw water. Water access challenges in urban areas had to do with irregular supply and cost. The water quality was sometimes poor. In terms of access to sanitation services, a participant had this to say: *“Getting a place of convenience is a challenge. It’s a public one and you will have to squat. It can be challenging. It is dirty and smelly. We pay to use it too (Thomas, M, GAR).”* Older rural residents also lamented walking longer distances at the mercy of the weather and snakes to openly defecate. A participant noted it was a common occurrence for them to improvise with containers during such periods and dispose of it later. 

#### 4.2.7. Caregiving and Associated Burden

Participants raised concerns about providing care for some household members especially grandchildren left behind, lamenting that it takes a heavy toll on their health and finances. Caregiving was highly gendered with older women in our sample much more involved than older men. Notably caregiving activities included feeding, bathing, clothing, and payment of school fees. Whereas some migrant children remit to support participants care for people left behind, such remittances are often irregular and inadequate. An old woman who cared for three grandchildren described it as the worst experience in her life. In her words.


*“When you are young and energetic you have no problem. Even 10 children can be taken care of. But at this stage it’s a big punishment. Two of them cannot bath by themselves. I have to cook for them. Paying their fee is also a problem. I can’t sleep at night and I wonder what will happen to them when I die (Hawa, F, BER).”*


## 5. Discussion 

Ghana’s older population has witnessed a steady increase over the past decades in the face of considerable social, economic, and environmental transformations [3,8,47]. However, there has been very little attention to the circumstances of older people in this context, particularly how their everyday experiences are situated within the broader geographies of health. This study contributes to the discourse on the “geographies of older people” by exploring the everyday experience of older people in Ghana. Our findings fit within the literature on “geographies of older people” and confirms the enduring view that older adults’ experiences are complex, gendered, and geographically variable [27,48,49,50]. Findings from this paper demonstrate the need to contextualize the experiences of older people within broader social, economic, political, and environmental settings.

At the microsystem level, we discovered that participants face challenges with instrumental activities of daily living particularly meal preparation, drawing water, household cleaning, shopping, and toileting. These challenges can be attributed to a decline in their functional ability and health status, conditions which were overwhelmingly reported by study participants. Earlier studies have reported similar challenges among older adults [4,27]. The findings also revealed gender disparities in participants’ experiences. In all situations except for performing most of the instrumental activities of daily living, older women appeared worse off. Older women mainly remained restricted to household spaces, performing various domestic duties such as caregiving, cooking, and cleaning. They also were discriminated against and abused more than their male counterparts. This is in line with the findings by [50] that the situation of older women in Ghana is characterized by poverty, and abuse. Explanations for these disparities can be found in the lower economic status of women and discriminatory cultural norms that ascribe women domestic roles in Ghana. Even though poverty is widespread, urban dwellers appeared to carry the heaviest burden, possibly due to the relatively higher cost of living in urban areas compared to the rural areas. Consistent with other studies poverty was found to negatively affect the mental and physical health of older people [3,16].

At the mesosytem, which encompasses the system of microsystems, we found mixed experiences among participants in the support they receive and their social status. An overwhelming majority of participants lacked social support despite the centrality of social support to the wellbeing of individuals [11]. This is concerning in the context of Ghana where formal support programs are limited. Earlier research in Ghana and related settings have made similar observations, attributing the lack of support for older people to a number of interrelated factors including weakening social ties, changing household structures, declining family sizes, economic challenges, and migration of adult children [7,15,27]. In contrast, a few participants reported receiving assistance, which may be due to their access to supportive social networks. In line with previous studies, some participants also carried the burden of providing care for household members, especially grandchildren [3,7]. Participants’ place of residence was also found to influence their overall experiences. Rural dwellers had more access to social support than their urban counterparts, suggesting that stronger social ties are more likely in rural than urban areas. More so, values that emphasize filial piety might be stronger in rural areas than urban areas. However, participants from urban areas and the Greater Accra Region had better access to health care services, food, and sanitation services. Better infrastructural development coupled with better economic conditions in urban areas in Ghana possibly explain these experiences [37]. These findings highlight the role of space in constructing aging experiences and feed into broader discourse on geographies of rural areas. 

At the exosystem, there was consensus that the central government has the duty to provide support services for older people. In line with this expectation, the government of Ghana introduced a number of pro-poor initiatives to improve the wellbeing of older people. Similar to many anti-poverty initiatives in Ghana, we found that these programs made minimal impact in improving the economic status of older people. Very few participants reported receiving financial assistance under the LEAP program. Likewise, not only are people under the age of 70 ineligible for free enrolment but also a lot of health problems faced by older people were not covered by the NHIS. This finding corroborates earlier studies on the topic in Ghana [3,17,51]. The effects of climate change on farming systems emerged as an important concern for participants especially those in rural areas and the Upper West and Bono East Regions where livelihoods heavily depend on agriculture. This finding is consistent with previous studies that have highlighted the negative impacts of changing climate on livelihoods in Ghana [52,53]. Other reported challenges at the exosystem level are the lack of health care facilities, water and sanitation services, and economic opportunities, which adversely affect participants’ wellbeing and quality of life [6,16]. 

Participants experiences at the macrosystem center on cultural norms and beliefs. Cultural norms and beliefs give meaning to aging and influence relationships and experiences of older people. These beliefs and norms also frame interactions within other domains of the environment. In line with prevailing cultural norms, older women expressed concerns about the lack of autonomy and access to communal resources. Some women farmers had challenges accessing farmlands. Women were commonly accused of witchcraft. We also noted that participants’ willingness to seek support, especially from non-kin members, was influenced by their world views. 

This study has a number of strengths and limitations. Theoretically, the study demonstrates the applicability of ecological systems theory and perspectives from the “geographies of older people” in investigating the experiences of older people. Through this we were able to illustrate how various individual- and structural-level factors interact to produce the everyday experiences of older people. The findings also augment the literature on older people in Ghana and more broadly in the African sub-region where research attention to older people is minimal. The use of sharing circles together with interviews allowed for the collection of data in a more in-depth, natural, and culturally sensitive manner. This study is among the first to collect data using sharing circles in the context of a developing country. With regards to limitations, the findings are a presentation of the dominant subjective views of a small sample of older people (urban dwellers, rural dwellers, women, and men) and hence, should be interpreted with caution. Also, in this study like most qualitative studies, there is the possibility of the meaning being lost during the translation of the results. We however minimized the possibility of such biases by involving several people fluent in the local languages to transcribe and validate transcripts. 

## 6. Conclusions 

This paper draws on the “geographies of older people” literature and ecological systems theory to illustrate how older people’s individual characteristics intersect with their social and physical environments to construct and give meaning to their lived experiences. We conclude that the experiences of older people in Ghana are complex and produced by social, economic, environmental, and political systems. Future studies can contribute to knowledge by drawing from geographic perspectives to explore specific concerns of older people, such as food insecurity, health services access, economic insecurity, and caregiver migration and its impacts. The findings demonstrate the need to scale-up existing social protection initiatives (the NHIS and LEAP program) to meet the needs of older people. This can be achieved by making adequate budgetary allocations to these funds and relevant institutions. Also, the eligibility criteria for these programs needs to be revised to cover all older people in need. Community-based social protection programs may also be instrumental to enhancing the wellbeing of older people. The provision of geriatric care services needs to be incorporated into the public health system and more health professional trained to provide such services. Given their heavy dependence on social networks, it would be beneficial to build stronger social and community networks to support older adults. Efforts should also be made to improve the economic status of older people. For instance, given the increasing life expectancy in Ghana, age of retirement could be revised to enable individuals over 60 years, capable and willing to work to remain in the labor force. There is also the need for the disbarment of existing discriminatory cultural norms that disadvantage older women. Although efforts have been made by the central government over the years to improve the wellbeing of older people in Ghana, our findings reveal they still face numerous challenges. Thus, promoting the wellbeing of older people must still be a top priority if Ghana is committed to meeting the 2030 agenda for sustainable development. 

## Figures and Tables

**Figure 1 ijerph-18-02337-f001:**
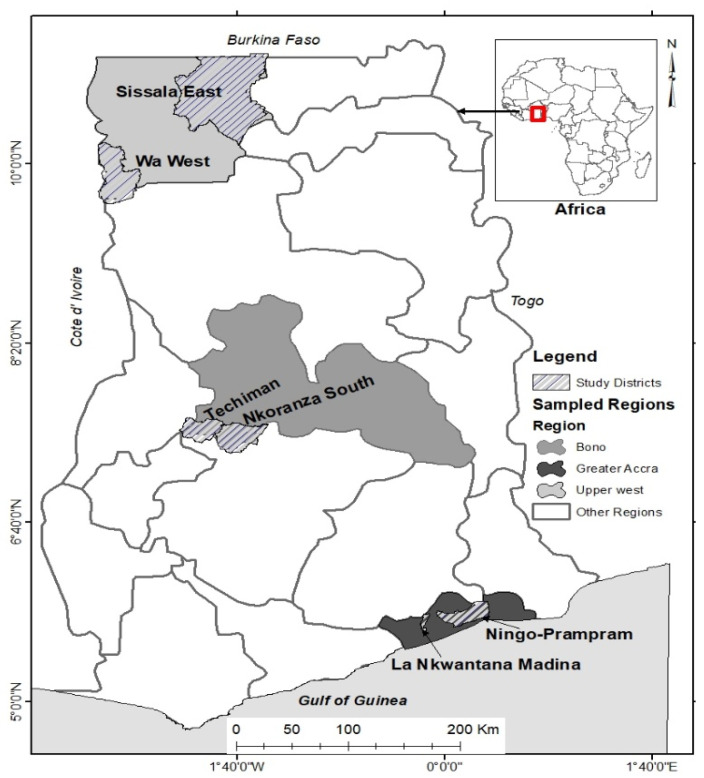
Study sites.

**Table 1 ijerph-18-02337-t001:** Participants characteristics―interviews (*n* = 42).

Characteristics	Number of Participants		
	Upper West Region (*n* = 19)	Bono East Region (*n* = 10)	Greater Accra Region (*n* = 13)
**Gender**			
Female	10	5	7
Male	9	5	6
**Age**			
60–69	10	6	4
70–79	8	3	7
80–99	1	1	2
**Highest school attended**			
None	13	7	2
High/Middle school	2	0	3
College/University	4	3	8
**Location**			
Rural	11	6	4
Urban	8	4	9
**Household size**			
5 or less	7	5	9
6 to 10	9	2	4
11 or more	3	3	0
**Number on pension**	6	3	10

**Table 2 ijerph-18-02337-t002:** Participants characteristics― sharing circles (*n* = 10).

Characteristics	Number of Participants	
	Upper West Region (*n* = 43)	Bono East Region (38)
**Gender**		
Female	22	21
Male	21	17
**Age**		
60–69	30	24
70–79	11	13
80–99	2	1
**Education**		
No education	37	28
Senior high school and less	2	1
College/University education	4	9
**Location**		
Rural	26	25
Urban	17	13

## Data Availability

Not Applicable.

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
