# Peer review of "“They Do Not Care about Us Anymore”: Understanding the Situation of Older People in Ghana"

_ijerph, 2021, doi:10.3390/ijerph18052337_

Round 1

Reviewer 1 Report

It is refreshing to see a paper that explores this unique and important topic.  This manuscript incorporates an ecological systems theory to describe the lived experiences of older people in Ghana.  The authors assert a need for this study by stating that though research exists on the challenges facing older adults in Ghana and much of sub-Saharan Africa, there is lack of studies on the lived experiences of this group within the broader context of public health and health geography.  The global increase in the older population, the accompanying levels of elevated morbidity and functional impairment, related psychosocial stressors, and the lack of engagement on this topic within sub-Saharan Africa further illustrate a need for this study. Examining this topic from the angle of geographical inquiry brings a distinct perspective to the paper.  Strengths include that the authors report that this is the first attempt to incorporate ecological systems theory and geographical inquiry to examine experiences among older people in Ghana. 

In lines 100-123, there is an insightful explanation of the five levels of the human environment (the microsystem, the mesosystem, the exosystem, the macrosystem and the chronosystem), which may be particularly for the helpful for the lay reader.

Lines 252-253 state “Interviews were conducted face-to-face at various public spaces in Twi, Sissali, Dagaare, Brifo, Ga or English depending on the preference of the individual”.  What were the different preferences of individuals?  Was the public space a facilitator or deterrent to participation?  Also, were incentives provided for participants to participate?  In the case where there were no monetary incentives, were there other considerations, such as providing refreshments or the like to acknowledge people’s time?

The findings from the participant interviews are compelling and profound.  The conclusions and recommendations are thorough and with good strategies, such as expanding existing social protection programs to help provide for older people.

Overall, this is an interesting, novel, and insightful study.  It is well-written, comprehensive, and well-researched.  There are a few clarifying questions, including about the intervention design, that the authors can explain.  Otherwise, this is a very sound paper.

Author Response

Thank you very much for your compliment. We are very grateful to you for agreeing to review this manuscript and your insightful suggestions. 

Responses:

  1. Regarding lines 252-253, participants generally preferred their homes which we considered public. We did not interview any participant in their bedrooms as we did not deem it ethical. We ensured that these environments where the interviews took place were comfortable and so participation was facilitated. We have made that clear in the manuscript (see lines 267-270).
  2.  For our intervention design, we have revised them to make them more clearer, starting from line 645 to 666. 

Thank you very much.

Reviewer 2 Report

A fabulous study and excellent written account of it. 

I only have a few minor comments for consideration and perhaps editing:

  • the first sentence in the abstract could be deleted, or should be revised, as most developed countries now have a rapidly aging population, and already high population proportion aged 65+.
  • some minor editing is needed throughout,
  • sorry, but it may be good to have a few additional references to published reports of relevant studies conducted in Ghana. Ghana is becoming a primary source of research information/evidence for Africa and for developing countries in general.  

Author Response

Thank you very much for reviewing this manuscript. Your comments are very insightful and helpful. Our responses are as follows:

  1. We have deleted the first sentence of our abstract as you recommended.
  2. We have also edited the entire manuscript as you suggested.
  3. We appreciate your suggestion on adding more published work from Ghana. We agree there is a lot of published work in Ghana on older people and so we have made efforts to include all relevant materials in our opinion. 

Once again thank you very much for making time to review our manuscript. 

Reviewer 3 Report

This is a very outstanding work, with methods that the authors clearly master, a profound knowledge of the relevant bibliography and a very clear writing. Congratulations to the International Journal of Environmental Research and Public Health for having received this original.

To improve it if it's possible, it should be mentioned in the discussion some advantages and disadvantages of purposive sampling.

  • Because the researchers oversee the selection process, their perspectives can influence the data they collect. They may unconsciously manipulate the data that is available to create outcomes that support their preconceived notions.

  • The only inference possibilities apply to the specific group that you are studying.

  • What was the percentage of all potentially eligible units that do not have responses to -at least a certain proportion of- the items in a survey questionnaire?

Author Response

We thank you very much for you compliment and making time to review this manuscript. Your comments are very insightful. See below our responses

  1. Like every other data collection technique, purposive sampling has advantages and disadvantages. Putting them in the main discussion however in our opinion will disrupt the flow. And so for the advantages, we present that as justification for using this technique in our methods sections (see lines 246 - 249). We also presented the disadvantage(s) as a limitation in our limitation paragraph.
  2.  We appreciate you question on the percentage of all potentially eligible units that do not have responses to - at least a certain proportion of- the items in a survey questionnaire. We like to clarify that we used an interview guide with a limited number of open ended questions. For this method of data collection, the emphasis is not on the units that do not have responses but rather depth and breadth on an issue.

Once again thank you very much.